# Assessing the Efficacy of Early Therapies against SARS-CoV-2 in Hematological Patients: A Real-Life Study from a COVID-19 Referral Centre in Northern Italy

**DOI:** 10.3390/jcm11247452

**Published:** 2022-12-15

**Authors:** Marta Colaneri, Teresa Chiara Pieri, Silvia Roda, Alessandra Ricciardi, Manuel Gotti, Jacqueline Ferrari, Luca Arcaini, Sara Rattotti, Antonio Piralla, Federica Giardina, Guglielmo Ferrari, Paolo Sacchi, Valentina Zuccaro, Fausto Baldanti, Raffaele Bruno

**Affiliations:** 1Division of Infectious Diseases, Fondazione IRCCS Policlinico San Matteo, 27100 Pavia, Italy; 2Division of Hematology, Fondazione IRCCS Policlinico San Matteo, 27100 Pavia, Italy; 3Department of Molecular Medicine, University of Pavia, 27100 Pavia, Italy; 4Microbiology and Virology Department, Fondazione IRCCS Policlinico San Matteo, 27100 Pavia, Italy; 5Department of Medical, Surgical, Diagnostic and Pediatric Science, University of Pavia, 27100 Pavia, Italy

**Keywords:** COVID-19, early remdesivir, molnupiravir, ritonavir-boosted nirmatrelvir, sotrovimab, hematological patients, hospitalizations rate, prolonged viral shedding

## Abstract

Early therapies to prevent severe COVID-19 have an unclear impact on patients with hematological malignancies. The aim of this study was to assess their efficacy in this group of high-risk patients with COVID-19 in preventing hospitalizations and reducing the SARS-CoV-2 shedding. This was a single-center, retrospective, observational study conducted in the Fondazione IRCSS Policlinico San Matteo of Pavia, Northern Italy. We extracted the data of patients with hematologic malignancies and COVID-19 who received and did not receive early COVID-19 treatment between 23 December 2021, and May 2022. We used a Cox proportional hazard model to assess whether receiving any early treatment was associated with lower rates of hospitalization and reduced viral shedding. Data from 88 patients with hematologic malignancies were extracted. Among the patients, 55 (62%) received any early treatment, whereas 33 (38%) did not. Receiving any early therapy did not significantly reduce the hospitalization rate in patients with hematologic malignancies (HR 0.51; SE 0.63; *p*-value = 0.28), except in the vaccinated non-responders subgroup of patients with negative anti SARS-CoV-2 antibodies at the time of infection, who benefited from early therapies against SARS-CoV-2 (HR 0.07; SE 1.04; *p*-value = 0.001). Moreover, no difference on viral load decay was observed. In our cohort of patients with hematologic malignancies infected with SARS-CoV-2, early treatment were not effective in reducing the hospitalization rate due to COVID-19, neither in reducing its viral shedding.

## 1. Introduction

Patients with hematological malignancies or who underwent hematopoietic stem-cell transplantation (HSCT) are considered at high risk of developing severe COVID-19 [1]. COVID-19-related mortality in patients with hematologic malignancies is higher than in the general population, being approximately 30% in several studies performed both in the pre- and in the post-vaccine era [1,2]. 

These patients are at higher risk of severe COVID-19, due to the long-lasting immunodeficiency resulting from malignancy itself, anticancer treatments, or HSCT [3,4]. Moreover, there is evidence of an impaired humoral immune or cellular response after anti-SARS-CoV-2 vaccination among patients with hematologic malignancies and HSCT patients [5], and a lower post-vaccination immunogenicity [6].

Furthermore, patients with hematologic malignancies and HSCT patients may have a prolonged viral shedding [7] compared to the roughly 10-days average duration usually reported for the general population [8]. Hence, plenty of studies have demonstrated a prolonged shedding duration of active virus, up to months after symptom onset [9,10,11,12]. 

Currently, there are valid options for symptomatic outpatients with COVID-19 that are at a high risk for progression to severe disease. Among those, the oral combination of nirmatrelvir/ritonavir is the recommended option [13], since it has been shown to reduce the risk for hospitalization by 89% [14]. Remdesivir has a similar efficacy and is an alternative option, but its use is impractical in some outpatient settings since it requires parenteral administration over 3 days [15]. A third option is anti-SARS-CoV-2 monoclonal antibodies which have variable activity against the different SARS-CoV-2 variants. Among them, Sotrovimab was the only one that retained some activity against BA.1/BA.1.1 sub-lineages of the Omicron variant [16], but is currently no longer effective against BA.2 [17]. Molnupiravir is another possible option. However, since its lower efficacy, which was roughly 30% in reducing COVID-19-related hospitalization by 28 days [18], the COVID-19 Treatment Guidelines Panel recommended its use only when the other options are contraindicated [19]. Together with COVID-19 related hospitalization and mortality rate reduction, these drugs might also lead to a significant reduction in viral load [20].

Although clinical trials generally exclude patients with hematologic malignancies, the European Conference on Infections in Leukemia recently recommended treating patients with hematologic malignancies with mild COVID-19 with these drugs [21].

The aim of this study was to assess the impact of early therapies in reducing the hospitalization rate and the 28-days mortality due to COVID-19 in patients with hematologic malignancies in our Hospital Fondazione IRCCS Policlinico San Matteo in Pavia, Northern Italy. We also aimed to evaluate the time length of viral shedding in patients with hematologic malignancies and HSCT patients who were and were not treated with early therapies.

## 2. Materials and Methods

### 2.1. Study Design

This study was a retrospective, single-center analysis of patients with a confirmed diagnosis of COVID-19 referred to our hospital. The study was approved by our Institutional Review Board (n.prot.0031226/22).

The medical records of all the adult patients with hematologic malignancies who tested positive for real-time reverse-transcription polymerase chain reaction (RT-PCR) from nasal swabs for SARS-CoV-2 and were consequently evaluated for early treatment in our clinic, were anonymized and abstracted on standardized data collection forms. In particular, patients suffering with myeloma, Hodgkin and non-Hodgkin lymphoma, chronic and acute leukemia, paroxysmal nocturnal hemoglobinuria, amyloidosis, and myelodysplastic syndrome/myeloproliferative neoplasms were included.

Only patients with mild to moderate COVID19 diseases were considered eligible for a therapy. Specifically, they did not present with any of the following features: oxygen saturation of <94% on room air; respiratory rate of >30 breaths/min; PaO_2_/FiO_2_ < 300 mmHg; and lung infiltrates > 50%.

We only extracted the data of patients evaluated between 23 December 2021 and 30 of April 2022, when the vast majority of COVID-19 cases were due to the Omicron variant.

The following exclusion criteria were applied: patients hospitalized for COVID-19 and/or requiring oxygen therapy for COVID-19 at the first clinical evaluation; asymptomatic patients.

### 2.2. Study Setting

One of the Infectious Diseases outpatients’ clinics of our hospital was allocated to the early treatment of COVID-19 outpatients from 23 December 2021. In this clinic, an infectious disease (ID) specialist was in charge of receiving daily e-mails from general practitioners and specialists of other units who promptly notified the cases of SARS-CoV-2 positive high-risk patients, both outpatients and patients admitted for reasons other than COVID-19.

The appropriate therapy for each notified patient was chosen by the ID specialist, according to both the inclusion and exclusion criteria and the availability of each drug’s pilot sheet. After signing an informed consent form, the patient was then examined and informed about the selected therapy.

Among these, ritonavir-boosted nirmatrelvir was selected as the first oral medication, but it was available only from 20 February 2022. If an intravenous (IV) drug was selected, remdesivir was administered as an IV infusion over 30 min at the recommended dosage of 200 mg for the loading dose on day 1, followed by a 100 mg maintenance dose administered on days 2 and 3. As regard with sotrovimab, it was given as a single 500 mg IV infusion, but it was used from the arrival of the Omicron BA.2 subvariant, at the end of April 2022. Patients were monitored during each infusion and observed for at least one hour after for signs and symptoms of hypersensitivity.

As a last resort, molnupiravir was administered to the individuals who were not eligible to any other drug.

### 2.3. Patients’ Characteristics

The demographic data included sex and age. Clinical data included symptoms at presentation, comorbidities (history of cancer, heart disease, hypertension, diabetes, chronic kidney disease, lung disease, and obesity), vaccination status, and anti-spike IgG antibodies for SARS-CoV-2 (results greater than or equal to the cut-off value 50.0 AU/mL were reported as positive). Type of hematological disease; ongoing chemotherapy; type and time of HSCT if performed.

The Italian Agency of Drugs (AIFA)s guidelines for excluding patients from one treatment rather than another was strictly followed.

### 2.4. SARS-CoV-2 RNA Detection

Total RNA was extracted on the MGISP-960 automated workstation using the MGI Easy Magnetic Beads Virus DNA/RNA Extraction Kit (MGI Technologies, Shenzhen, China). Detection of SARS-CoV-2 RNA was performed using the SARS-CoV-2 variants ELITe MGB^®^ kit (ELITechGroup, Puteaux, France; cat. no. RTS170ING) on the CFX96 Touch Real-time PCR detection system (BioRad, Mississauga, ON, Canada).

### 2.5. Outcomes

The primary outcome was to evaluate the impact of early therapies, such as remdesivir, molnupiravir, ritonavir-boosted nirmatrelvir, and sotrovimab, in preventing the hospitalization due to COVID-19 of patients with hematologic malignancies infected by SARS-CoV-2 by day 28.

In particular, we considered the progression of COVID-19 as the presence of clinical manifestations which are consistent with the categories of moderate, severe, and critical illness defined by the National Institute of Health Guidelines [22].

We also evaluated admission to the intensive care unit (ICU) of our hospital and the intra-hospital mortality by day 28.

The secondary outcomes were to evaluate the effect of the single drug in preventing the 28 days hospitalization due to COVID-19, to evaluate the length of SARS-CoV-2 viral shedding of patients receiving early therapies versus those who did not receive them, and finally, to evaluate the impact of the early therapies in patients with hematologic malignancies with negative SARS-CoV-2 antibodies at the time of evaluation.

### 2.6. Statistical Analysis

Data for continuous variables were presented as means and standard deviations.

Categorical variables were presented as frequencies and percentages. Comparisons between the treated and non-treated groups of patients with hematologic malignancies were performed using chi-square tests for categorical variables and Mann–Whitney tests for non-normal continuous data.

The log-rank test was used to estimate the difference between the 28-day Kaplan–Meier hospitalization curves of patients who received and did not receive early therapies. The duration of viral shedding was calculated by using the Kaplan–Meier curves and tested by the log-rank test for survival curve comparison. When viral clearance could not be determined, the duration was censored with the last positive sample. A Cox proportional hazard model was performed controlling for sex, age, number of underlying comorbidities, and number of anti-SARS-CoV-2 vaccinations performed. A multivariable Cox proportional-hazard regression model was also performed to evaluate the impact of each drug on the hospitalization rate compared to no drugs. 

Finally, a multivariable Cox proportional-hazard regression model was performed to evaluate the impact of early therapies in patients with hematologic malignancies with negative anti SARS-CoV-2 antibodies at the time of evaluation.

The results were reported as hazard ratios (HRs) and 95% confidence intervals (CIs). Statistical analyses were conducted using R (version 4.1.2).

## 3. Results

Data from 88 patients were extracted. A total of 55 (62%) received early therapy and 33 (38%) did not. Demographic, clinical, and treatment characteristics are presented in Table 1.

Most patients were vaccinated against SARS-CoV-2 (94%). However, among them, only 44 (50%) patients had positive IgG anti- SARS-CoV-2 spike protein.

Regarding the treatment, 55 (62%) patients received an early treatment for SARS-CoV-2. Fifteen (27%) were treated with remdesivir, 10 (18%) with ritonavir-boosted nirmatrelvir, 15 (27%) with sotrovimab, and 15 (27%) with molnupiravir.

Globally, the length of PCR positivity for SARS-CoV-2 on nasal swab had a mean of 26.3 (±21.6) days, 25.4 (±18.0) and 27.7 (±24.0) for the treated and untreated group, respectively. Among the treated patients, six (11%) developed COVID-19 related pneumonia, with five of them requiring oxygen therapy and hospitalization. None of the treated patients required ICU admission. Moreover, six untreated patients were hospitalized for COVID-19 related pneumonia. Among them, one was admitted to the ICU, while two died.

### 3.1. Impact of Early Therapies on the Outcomes

Regarding our primary outcome, treatment with any considered early therapy did not significantly reduce hospital admission by 28 days (Figure 1).

Similarly, after accounting for potential confounders, the multivariable Cox proportional-hazard regression model showed that an early treatment with any of the considered drugs did not significantly reduce the hospitalization rate (HR: 0.51; SE 0.63; *p* = 0.28) (Table 2).

Additionally, the multivariable Cox proportional-hazard regression model showed that none of the early treatments did significantly reduce the hospitalization at day 28 compared with no treatment (Table 3).

Finally, the multivariable Cox proportional-hazard regression model showed that patients with hematologic malignancies with negative anti SARS-CoV-2 antibodies at the time of infection were at a significantly increased risk of hospitalization if not treated in a timely fashion with early therapies.

Specifically, after accounting for sex, age, number of vaccinations, and comorbidities, being untreated was significantly associated with an increased risk of hospitalization among patients with hematologic malignancies with negative anti SARS-CoV-2 antibodies (Table 4) (Figure 2).

### 3.2. SARS-CoV-2 RNA Load Kinetics

In a subset of patients (49/79; 62.0%), the duration of viral load was available, and the median duration was 15 days (range 8–87 days) for untreated, 21 days (range 8–31 days) for Remdesivir, 17 days (6–46 days) for sotrovimab, and 17 days (8–27 days, log-rank test *p* = 0.48) for molnupiravir (Figure 3A). Only one patient treated with ritonavir-boosted nirmatrelvir had data on viral load duration (8 days censored). Among the untreated group, the more prolonged infection was observed in a patient with RNA detected at 87 days after first positivity, while in the treated patients’ group, the more prolonged shedding was observed in one case treated with Sotrovimab with detectable RNA at 52 days after first positivity.

In addition, in a subset of patients (43/79; 54.4%) Ct values were available and used to calculate viral load decay normalized per day (Ct/day). No difference in viral load decay was observed between the groups of patients. However, the highest reduction in SARS-CoV-2 RNA was observed in untreated patients (median 1.27, range 0.50–3.25 Ct/day) as compared to Remdesivir (median 0.78, range 0.40–1.60 Ct/day), sotrovimab (median 0.75, range 0.29–2.22 Ct/day) and Molnupiravir (median 1.00, range 0.61–1.88 Ct/day) (Figure 3B). Only one patient treated with ritonavir-boosted nirmatrelvir had data on viral load decay (2.13 Ct/day).

## 4. Discussion 

In the present study, we did not notice a significant impact of early anti-SARS-CoV-2 treatments on the COVID-19-related 28-day hospitalization rate and SARS-CoV-2 load decay in patients with hematological malignancy or HSCT. However, untreated patients with negative anti-SARS-CoV-2 antibodies had a significantly higher risk of being hospitalized than treated ones.

Patients with hematologic malignancies and HSCT might experience a relatively slow viral decay and, as a result, the duration of RT-PCR positivity in these patients was longer than that of other patients [7]. Based on previous studies, a beneficial impact of early therapies on hastening the SARS-CoV-2 viral decay was expected [14,23,24]. Interestingly, our data did not confirm this hypothesis. This result should be taken with caution since the absence of a significant effect could also be explained by a lack of statistical power due to the relatively small sample size Although a prolonged duration of RT-PCR positivity does not indicate higher severity of COVID-19 [24], the fact that the viral load in these patients is long-lasting has serious healthcare implications. In fact, RT-PCR positivity in these patients generally prevents the implementation of specific treatments for their underlying disease, and access to outpatients’ care services.

In summary, the clinical and therapeutic management of hematologic malignancies and HSCT represent a major challenge for physicians. In this regard, and especially because of the constant surfacing of new SARS-CoV-2 variants of concern, we should reflect on the need of patients with hematologic malignancies or HSCT for updated vaccination strategies, such as prompt additional vaccine doses, which might be an effective choice to enhance immunity response [25]. Even though it has been reported that the severity of the Omicron SARS-CoV-2 variant is attenuated [26,27], this is likely due to population immunity rather than to a characteristic of the virus. Therefore, despite the ongoing trend of gradually relaxing epidemic containment measures, these patients should be instructed to maintain infection control measures, such as aerosol and contact full isolation, social distancing, and wide use of masks and personal hygiene measures.

We believe that it is extremely valuable to perform real-life studies on these patients, because of their high risk of mortality and morbidity due to COVID-19 [28,29,30], and their low response to anti-SARS-CoV-2 vaccines [6] due to their specific illness, chemotherapy, and other immunosuppressive treatments. Our data confirm this unfortunate trend, as only slightly more than half of the subgroup of fully immunized patients with hematologic malignancies were serologically positive for IgG anti-SARS-CoV-2 spike protein. The fact that patients with hematologic malignancies who have failed to mount an adequate SARS-CoV-2 vaccine response encounter poor outcomes is well known [26], and our data support the relevance of providing a timely treatment to these patients using early therapies against COVID-19.

We have to mention some limitations of our study, such as its retrospective and monocentric nature, and the relatively small sample size. Moreover, due to the real-life experience, we did not exclude those patients treated with molnupiravir, which is less effective than the other treatments [18]. Finally, since our sample only includes patients who were infected by the Omicron variant, the generalization of our results to patient affected by other variants should be executed with caution. However, to the best of our knowledge, no previous data supporting the use of early drugs in patients with hematologic malignancies or HSCT are available. Therefore, we believe that this study fills this literature gap with real-life daily practice findings.

In conclusion, we believe that reporting these real-life data may still be the most appropriate approach to appreciate how to focus our full consideration of patients with hematologic malignancies and HSCT patients from different perspectives. However, more data are needed to understand the best way to manage the SARS-CoV-2 infection in this particularly fragile population.

## Figures and Tables

**Figure 1 jcm-11-07452-f001:**
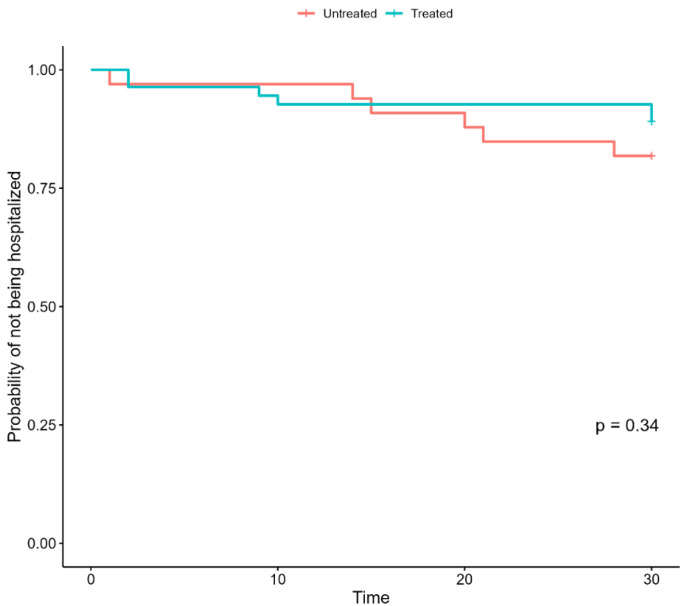
Kaplan–Meier curves of hospitalization in untreated and treated patients with hematologic malignancies and HSCT patients.

**Figure 2 jcm-11-07452-f002:**
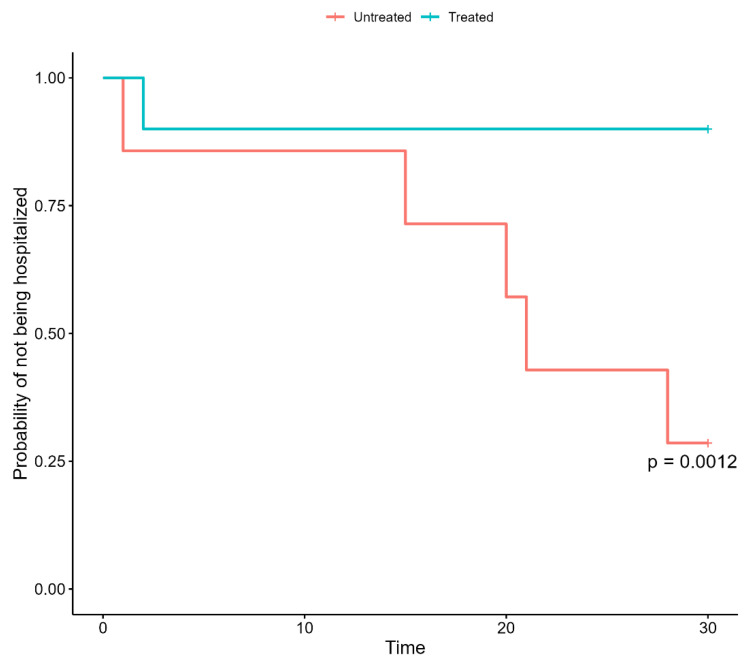
Kaplan–Meier curves of hospitalization in untreated and treated patients with hematologic malignancies and HSCT patients with negative anti-SARS-CoV-2 antibodies.

**Figure 3 jcm-11-07452-f003:**
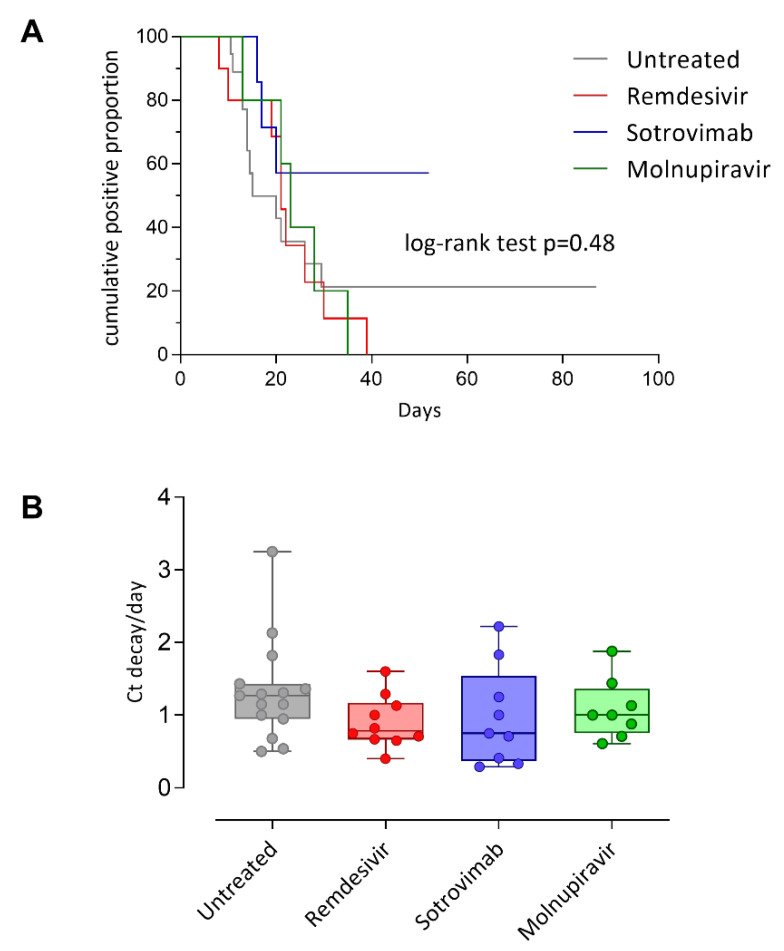
Kaplan–Meier curves of viral shedding duration in untreated and treated patients (**A**). SARS-CoV-2 RNA load clearance in different patients’ categories (**B**).

**Table 1 jcm-11-07452-t001:** Demographic and clinical characteristics.

		All Patients (88)	Treated (55)	Non-Treated (33)	*p*-Value
**Sex, *n* (%)**	Female	47 (53)	27 (31)	20 (23)	
	Male	41 (47)	28 (32)	13 (15)	0.41
**Age, Median (IQR)**		63 (49.0, 71.2)	62 (52.5, 70.0)	63 (48, 72)	0.89
**Vaccination doses, Mean (sd)**		2.7 (0.7)	2.6 (0.8)	2.7 (0.5)	0.69
**Days from last vaccination, Mean (sd)**		124.1 (65)	128.1(64.3)	116.9 (67.1)	0.51
**Remdesivir, *n* (%)**		-	15 (27)		-
**Ritonavir-boosted Nirmatrelvir, *n* (%)**		-	10 (18)		-
**Sotrovimab, *n* (%)**		-	15 (27)		-
**Molnupiravir, *n* (%)**		-	15 (27)		
**Bone marrow transplantation, *n* (%)**		24 (27)	18 (75)	6 (25)	0.22
**Days from bone marrow transplantation, Mean (sd)**		1307.4 (1793.8)	1390.3 (1981.8)	1009 (929.2)	0.68
**Type of Bone marrow transplantation, *n* (%)**	Autologous	20 (22)	14 (25)	6 (18)	
	Allogenic	4 (4)	4 (7)	0 (0)	0.28
**Hematological disease, *n* (%)**	Myeloma	26 (29)	17 (31)	9 (27)	
	Hodgkin Lymphoma	8 (9)	3 (5)	5 (15)	
	High-Grade Non-Hodgkin Lymphoma	12 (14)	10 (18)	2 (6)	
	Acute Myeloid Leukemia	4 (4)	3 (5)	1 (3)	
	Low-Grade Non-Hodgkin Lymphoma	16 (18)	7 (13)	9 (27)	
	Chronic Lymphocytic Leukemia	4 (4)	3 (5)	1 (3)	
	Chronic Myeloid Leukemia	8 (9)	7 (13)	1 (3)	
	MDS/MPN	3 (3)	2 (4)	1 (3)	
	Paroxysmal Nocturnal Hemoglobinuria	1 (1)	0 (0)	1 (3)	
	Acute Lymphocytic Leukemia	4 (4)	2 (4)	2 (6)	
	Amyloidosis AL	1 (1)	1 (2)	0 (0)	0.25
**Immunosuppressive therapies, *n* (%)**	Rituximab	20 (23)	13 (24)	7 (21)	1.00
	Obinutuzumab	5 (6)	3 (6)	2 (6)	1.00
	Methotrexate	10 (11)	5 (9)	5 (15)	0.60
	CHOP	15 (17)	12 (22)	3 (9)	0.21
	CHOEP	1 (1)	0 (0)	1 (3)	0.79
	ABVD	4 (4)	1 (2)	3 (9)	0.29
	Poli chemotherapy (VCR, Ara-C, Ida, EDX, Cisplatin, Bendamustine)	21 (24)	13 (4)	8 (24)	1.00
	VD (Bortezomib-Dexamethasone)	12 (14)	8 (14)	4 (12)	1.00
	Eculizumab	1 (1)	0 (0)	1 (3)	0.80
	Tyrosine kinase inhibitors (TKIs)	13 (15)	10 (18)	3 (9)	0.37
	Others (Daratumumab, Isatuximab, IMIDs, Brentuximab, Ab anti-PD1-PDL1)	30 (34)	17 (31)	13 (39)	0.60
**Days between last therapy and examination, mean (sd)**		3205.2 (11,379.2)	2902.1 (10,844.7)	3799 (12,582.5)	0.75
**Positive anti SARS-CoV-2 antibodies, *n* (%)**		44 (50)	20 (36)	24 (73)	<0.01
**Viral decay (sd)**		26.3 (21.6)	25.4 (18.0)	27.7 (24)	0.63
**Comorbidities**	NPL, *n* (%)	59 (69)	32 (63)	25 (78)	0.22
	CKD, *n* (%)	8 (10)	3 (7)	5 (15)	
	CVD, *n* (%)	14 (16)	8 (15)	6 (18)	0.90
	HTN, *n* (%)	34 (39)	21 (39)	13 (39)	1.00
	DM, *n* (%)	10 (11)	5 (9)	5 (15)	0.62
	LD, *n* (%)	10 (11)	7 (13)	3 (9)	0.83
	HCV, *n* (%)	2 (3)	2 (4)	0 (0)	0.70
	Obesity, *n* (%)	1 (1)	0 (0)	1 (3)	0.80
	Smoke, *n* (%)	10 (13)	5 (12)	5 (16)	0.90
	Number of comorbidities, mean (sd)	1.5 (1.2)	1.4 (1.1)	1.7 (1.3)	0.24
**Mortality, *n* (%)**		2 (2)	0 (0)	2 (6)	0.27
**Hospital admission, *n* (%)**		12 (14)	6 (11)	6 (18)	0.52
**ICU admission, *n* (%)**		1 (1)	0 (0)	1 (3)	0.79
**Stay at Home, *n* (%)**		78 (87)	50 (91)	28 (85)	0.60
**Symptoms, *n* (%)**					
	Asymptomatic	10 (12)	1 (2)	9 (30)	<0.01
	Fever	39 (48)	30 (57)	9 (32)	0.06
	Cough	32 (39)	24 (45)	8 (29)	0.2
	Pharyngodinia	25 (31)	16 (30)	9 (32)	1.00
	Dyspnea	10 (13)	3 (6)	7 (25)	0.04
	Diarrhea	2 (2)	2 (4)	0 (0)	0.77
	Asthenia	15 (18)	10 (19)	5 (17)	1.00
**Pneumonia**		12 (14)	6 (11)	6 (21)	0.39
**Oxygen therapy**		11 (13)	5 (9)	6 (19)	0.33

**Notes**: MDS/MPN, Myelodysplastic syndrome/Myeloproliferative neoplasms; ABVD, Adriamycin/bleomycin/vinblastine/dacarbazine; VCR, vincristine; EDX, 4′-epidoxorubicin; IMIDs, immunomodulatory drugs; NPL, Neoplasia; CKD, Chronic Kidney Disease; CVD, Cardiovascular Disease; HTN, Hypertension; DM, Diabetes Mellitus; LD, lung disease; Obesity considered as Body Mass Index (BMI) > 30 kg/m^2^; ICU, intensive care unit; Positive anti-SARS-CoV-2 antibodies was considered when IgG anti-trimeric SARS-CoV-2 spike protein were ≥50 AU/mL; HCV, presence of antibodies against HCV. Data are reported as absolute number and percentage and mean with standard deviation.

**Table 2 jcm-11-07452-t002:** Multivariate Cox regression for 28-day hospital admission.

Variable	HR	SE	*p*-Value
Treatment	0.51	0.63	0.28
Sex	0.29	0.68	0.07
Age	1.01	0.02	0.73
Number of vaccinations	1.42	0.61	0.56
Comorbidities	1.63	0.26	0.06

**Table 3 jcm-11-07452-t003:** Multivariate Cox regression for 28-day hospital admission considering the impact of each treatment.

Variable	HR	SE	*p*-Value
Paxlovid	0.51	1.10	0.55
Remdesivir	1.16	0.71	0.83
Molnupiravir	0.28	1.09	0.24
Sotrovimab	0.24	1.09	0.19
Sex	0.32	0.62	0.07
Age	1.03	1.41	1.41
Number of vaccinations	1.43	0.56	0.57

**Table 4 jcm-11-07452-t004:** Multivariate Cox regression for 28-day hospital admission of patients with hematologic malignancies with negative anti SARS-CoV-2 antibodies.

Variable	HR	SE	*p*-Value
Treatment	0.07	1.04	0.001
Sex	0.37	0.98	0.31
Age	1.00	0.04	0.91
Number of vaccinations	1.05	0.74	0.93
Comorbidities	1.63	0.35	0.16

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
