# Peer review of "Assessing the Efficacy of Early Therapies against SARS-CoV-2 in Hematological Patients: A Real-Life Study from a COVID-19 Referral Centre in Northern Italy"

_jcm, 2022, doi:10.3390/jcm11247452_

Round 1
Reviewer 1 Report
Patients with hematological malignancies (HM) and hematopoietic stem-cell transplantation are usually excluded from clinical trials and therefore SARS-CoV-2 disease manifestation in this population is obscure. Here authors demonstrate the impact of early treatment in preventing hospitalization in such a population. Overall the study highlights the important observation that the early treatments do not reduce hospitalization rate in HM outpatients but vaccinated subgroups with detectable SARS-CoV-2 specific antibodies were observed to get the benefit and therefore appropriate therapies may be considered to treat the disease in such population. The study makes use of appropriate statistics to represent adequately matched populations.
However, there are a few minor concerns,
1. Figure 1, the legend says the rate of hospitalization however the Y-axis represents the probability of survival. It is the same case in Figures 2 and 3A, please clarify
2. Figure 3a, indicates virus shedding duration, please clarify for the untreated group if this duration is more than 80 days. Even for the treatment groups is it 40-60 days? Y-axis percent survival? Statistics?
3. For each figure legend please indicate if the population group is HM or HSCT or control.
Author Response
Attached is our response to the comments, which are perfectly congruent with our paper. Therefore, we thank you for the improvement in our work.

Reviewer 2 Report
This is an interesting, important and potentially good clinical paper but, unfortunately, I need to point out a potential pitfall/complication that needs to be addressed:
1) I noticed that the data were collected between Dec 2021 and Feb 2022. This was when Omicron has begun to spread throughout the world. It first struck South Africa in Nov 2021. What could complicated matters is that computational, experimentally and clinically studies have shown that Omicron is inherentlyattenuated and different from other variants:
https://pubmed.ncbi.nlm.nih.gov/34971823/
https://www.thelancet.com/journals/lancet/article/PIIS0140-6736(22)00017-4/fulltext
https://www.nature.com/articles/s41586-022-04479-6
https://www.mdpi.com/2218-273X/12/10/1353/htm#B16-biomolecules-12-01353
https://pubmed.ncbi.nlm.nih.gov/35625559/
https://pubmed.ncbi.nlm.nih.gov/34971823/
My concern is that the arrival of Omicron in Italy may have complicated the data. And multivariate analysis is very sensitive to such issue.
2) My question is then: Do the authors have data pertaining to the variant that the patients were infected with? If so, they need to redo the statistics using an additional variable.
3) If not, the authors need to pore over literature and scientific data to to address the prevalence of omicron and other variants during the period of Dec 2021 through Mar 2022 and to fully explain the deficiency of their study.
3)
Minor issue: The paper mentions the use of regression analysis. It would be interesting to see the r2, which is the coefficient of determination as this will tell much about the correlation
Author Response

(The authors gave the same response as above.)

Round 2
Reviewer 2 Report
There is some improvement in this version. However, I still think there is still room for further improvement. The fact that this case study is mainly about Omicron tells us that it is likely that this does not apply to other variants. I suspect that the reason that the drugs do not improve the conditions of the patients is that Omicron is inherently much milder. All these have not been sufficiently highlighted. It must also be clearly stated with references that Omicron is intrinsically milder. I think that these are very important points that have not been sufficiently addressed.
Author Response
As a first remark, we thank the reviewer for his comment, which we consider important for our paper.
As our study included almost all HM or HSCT patients infected with the SARS-CoV-2 Omicron variant of concern (VOC), it may be generalized by some other physicians worldwide, only for this specific VOC.
Therefore, we have added a statement to the discussion, which I hope will resolve the first part of the criticism posed.
In fact, it should be noted that this is precisely the VOC in current circulation, with which, therefore, it might be important to come to terms.
Secondly, let us consider the reviewer's second point, which appears to be critical of the milder clinical presentation of COVID-19 caused by the Omicron variant. This fact could have somewhat affected the efficacy of the early treatments analysed in our paper.
While what the reviewer asserts is certainly supported by a flourishing scientific literature that he himself cites, we believe that the SARS-CoV-2 Omicron variant’s milder outcomes are likely due to more population immunity rather than the virus’ properties.
This is even more true for patients who are not only more fragile than others due to their comorbidities, such as the category of HM and HSCT patients we have considered in this paper, but who are often not responders to the absolute most effective preventive strategy, which is the SARS-CoV-2 vaccination.
(ref: Omicron severity: milder but not mild Joshua Nealon and Benjamin J Cowling , Lancet (London england)(nih.gov))
Therefore, we consider it of great value to add the reviewer's observation of the limitations of our study to our text, but always bearing in mind that, since we are dealing with a particular category of patients with little available data, there is reasonable doubt that makes this argument more complex and varied than it might seem.
